# Therapeutic Effects of Apamin as a Bee Venom Component for Non-Neoplastic Disease

**DOI:** 10.3390/toxins12030195

**Published:** 2020-03-19

**Authors:** Hyemin Gu, Sang Mi Han, Kwan-Kyu Park

**Affiliations:** 1Department of Pathology, College of Medicine, Catholic University of Daegu, Daegu 42472, Korea; guhm1207@cu.ac.kr; 2National Academy of Agricultural Science, Jeonjusi, Jeonbuk 54875, Korea; sangmih@korea.kr

**Keywords:** apamin, Component of bee venom, SK channels

## Abstract

Bee venom is a natural toxin produced by honeybees and plays an important role in defending bee colonies. Bee venom has several kinds of peptides, including melittin, apamin, adolapamine, and mast cell degranulation peptides. Apamin accounts for about 2%–3% dry weight of bee venom and is a peptide neurotoxin that contains 18 amino acid residues that are tightly crosslinked by two disulfide bonds. It is well known for its pharmacological functions, which irreversibly block Ca2+-activated K+ (SK) channels. Apamin regulates gene expression in various signal transduction pathways involved in cell development. The aim of this study was to review the current understanding of apamin in the treatment of apoptosis, fibrosis, and central nervous system diseases, which are the pathological processes of various diseases. Apamin’s potential therapeutic and pharmacological applications are also discussed.

## 1. Introduction

Bee venom (*Apis mellifera* L.) has traditionally been used to treat a variety of diseases, including arthritis, back pain, cancerous tumors, and multiple sclerosis [1,2]. Bee venom contains various bioactive proteins, such as melittin, apamin, adolapin, phospholipases A2 and B, hyaluronidase, serotonin, histamine, dopamine, and noradrenaline [3]. The peptides melittin, apamin, and mast cell degranulating peptides are exclusive to bees.

Given the anti-inflammatory properties of bee venom, various forms of traditional bee venom treatment, including stings, venom injections, and venom acupuncture, have been used to alleviate pain and to treat chronic inflammatory diseases, such as rheumatoid arthritis and multiple sclerosis [2,4,5]. In addition, our previous studies suggested that bee venom attenuates atherosclerosis atopic dermatitis and periodontitis by the suppression of pro-inflammatory cytokines [6,7,8].

Moreno et al. [3] examined the most recent and innovative therapeutic and biological applications of the most widely known components of bee venom, namely melittin and apamin. Apamin, an 18 amino acid peptide neurotoxin, is one of the bioactive components of bee venom, making up 2%–3% of its total dry weight [1]. 

Apamin has long been known as a specifically selective blocker of small conductance Ca2+-activated K+ (SK) channels [9], therefore, it acts as an allosteric inhibitor [10]. These channels play a pivotal role in various pathophysiological responses, such as atherosclerosis, Parkinson’s disease, and hepatic fibrosis [11,12,13,14]. Kim et al. [15] found that high concentrations (≥0.5 μg/mL) of apamin increase pro-inflammatory cytokines. A relatively low concentration of apamin has not been shown to affect cell death. According to several studies, the proposed treatments use relatively low concentrations (1–10 μg/mL) of apamin injections [3,15]. Thus, it has been suggested that the therapeutic element does not have a toxicologic effect on target lesions. In addition, recent studies have demonstrated apamin’s biological and pharmacological functions [16]; however, little is known about the molecular mechanisms and pathogenesis involved in SK channel blockers or in the anti-inflammatory process. Therefore, this review focuses on an overview of a variety of studies on the pharmacological properties of apamin in atherosclerosis, liver fibrosis, central nervous system (CNS) disease, and anti-inflammatory responses. Given the importance of the pharmacological action of apamin against various problematic diseases, this review provides a thorough examination of apamin and summarizes its potential therapeutic mechanisms (Table 1).

## 2. Pharmacological Actions of Apamin

### 2.1. Ca^2+^ Channels Blocker

The SK channel consists of three members, SK1, SK, and SK3, which are voltage independent. SK channels link intracellular Ca2+ transients to changes of the membrane potential by promoting K+ efflux with increasing intracellular calcium during the action potential [11,22]. SK channels were first identified in the brain [23,24] and later described in various tissues, including smooth muscle, endothelial, epithelial, and blood cells [24,25]. 

SK1, 2, and 3 channels are widely expressed throughout the body, including the heart, the liver, and skeletal muscle [22]. Of these, SK1 and SK2 channels are co-expressed in the neurons of the neocortex, the hippocampal formation, the cerebellum, and brain stem, and SK3 channels have been reported to be present in the midbrain, hypothalamus, and hypothalamus regions [26,27,28].

Apamin has long been known as a selective blocker of SK channels [29], and it is a peptide with a highly specific mode of pharmacological action. Apamin binds to the pores of the SK channels, acts as an allosteric inhibitor, and blocks them. Subsequently, it suppresses delayed cell hyperpolarization [10]. This binding specificity provides apamin with its electrical properties. 

SK channels have been associated with learning regulation, and they may play specific roles in dopamine-related disorders [12]. SK channels are believed to affect learning by regulating synaptic plasticity through N-Methyl-d-aspartate (NMDA) receptors [30]. Based on these findings, it is suggested that apamin can selectively block SK cannels involved in learning and memory regulation, thereby contributing to learning and memory control [16,21,31].

### 2.2. Atherosclerosis

Atherosclerosis is a progressive disease in which the inner wall of the arteries forms plaque, consisting mainly of debris from cholesterol, other lipids, and cell death [32]. Many macrophages can be observed in atherosclerotic lesions, and the initial lesions of atherosclerosis are characterized by an infiltration of macrophages, a proliferation of smooth muscle cells, and the presence of macrophage-derived foam cells [33]. Macrophages, which are differentiated from blood peripheral monocytes, are mixed with modified lipoproteins and then transformed into lipid-rich foam cells, which comprise the main feature of atherosclerosis [34]. The accumulation of free cholesterol and oxidatively modified cholesterol induces macrophage apoptosis [35].

Macrophages are pluripotent inflammatory cells with the ability to synthesize and to secrete pro-inflammatory cytokines, such as tumor necrosis factor (TNF)-α, interleukin (IL)-1β, IL-8, and IL-6. These cytokine plays a central role during the development of atherosclerosis [36]. Pro-inflammatory cytokines are regulated at the level of transcription by several transcription factors, including nuclear factor kappa B (NF-κB) and activator protein-1 [37,38]. NF-κB is a transcription factor that affects different stages of the atherosclerotic process, including early atherosclerosis, foam cell formation, proliferation of smooth muscle cells, and fibrous cap formation [39,40,41].

Several studies [42,43,44] have shown that some calcium channel blockers can reduce atherosclerotic lesions, the production of oxidative stress, and the expression of inflammatory cytokines. Apamin, an SK channel blocker, has been reported to exert an anti-inflammatory effect with a decrease in seromucoid and haptoglobin levels [45].

Kim et al. [13] evaluated the anti-atherosclerotic mechanisms of apamin function in THP-1-derived macrophages and examined the anti-atherosclerotic effects of apamin in mouse models of atherosclerosis. Treatment with apamin in lipopolysaccharide (LPS)-treated THP-1-derived macrophages inhibited inflammatory responses due to a decrease in the NF-κB signal pathway. Similarly, some studies have demonstrated that apamin treatment effectively downregulates the NF-kB signaling pathway and signal transducers and activators of transcription (STAT) in vitro, thereby inhibiting pro-inflammatory cytokines and Th2 lymphocyte chemokines [13,15]. Kim et al. [13] showed that intracellular lipid levels are inhibited by apamin in oxidized low-density lipoprotein (oxLDL)-treated macrophages. The inhibition of macrophage activity attenuates the elevation of lipid levels in oxLDL-induced macrophage apoptosis through apamin. Furthermore, treatment with apamin for atherosclerotic mice predominantly alleviates serum lipids, Ca2+ levels, pro-inflammatory cytokines, adhesion molecules, fibrotic factors, and macrophage infiltration [13]. In the atherosclerosis model, mice injected with apamin had an inhibited expression of TNF-α, intracellular cell adhesion molecule (ICAM)-1, vascular cell adhesion molecule (VCAM)-1, transforming growth factor (TGF)-β1, fibronectin, and the NF-κB signaling pathway [13]. Apamin also suppresses platelet-derived growth factor subunit B homodimer (PDGF-BB)-induced vascular smooth muscle cell proliferation and migration [15]. 

The consequences of macrophage apoptosis may vary from early to late atherosclerotic lesions [46]. For early lesions, apoptosis appears to be efficient for cell phagocytosis, and macrophage apoptosis reduces lesion cellularity and lesion progression. For late lesions, a number of factors can contribute to phagocytic clearance defects in apoptosis macrophages, causing secondary necrosis and pro-inflammatory responses in these cells [47,48]. Pro-apoptosis processes that occur in atheroma include oxidant stress through high levels of cytokine activation of oxidized low-density lipoproteins (oxLDL) [49]. Apoptosis is thought to be caused by an extrinsic pathway using death receptors and/or by an intrinsic pathway, including cytochrome C and caspase activation [50]. OxLDL and oxidized lipids are also produced in atherosclerotic lesions [51].

Kim et al. [17] investigated the anti-apoptosis mechanism of oxLDL-induced THP-1-derived macrophage apoptosis. Apoptotic macrophages were significantly reduced by reducing the expression of pro-apoptotic genes, such as Bax, PARP, caspase-3, and cytosolic cytochrome C protein levels, through treatment with apamin. Likewise, treatment with apamin increased the expression of anti-apoptotic genes, such as Bcl-2, Bcl-xL, and mitochondrial cytochrome C activation. The authors demonstrated that apamin reduces apoptosis through the mitochondria-related apoptotic pathway. Furthermore, treatment with apamin in mouse models of atherosclerosis dramatically alleviated apoptotic cell death [17].

In summary, apamin inhibits inflammatory responses in macrophages and weakens the LPS/fat-induced atherosclerosis model in vivo. Apamin could potentially be used to develop new agents that inhibit macrophage apoptosis to protect against atherosclerosis. In the studies reviewed, apamin has been proposed as another therapeutic agent for the treatment and prevention of atherosclerosis. 

### 2.3. Ventricular Fibrillation

Heart failure is associated with structural and electrophysiological remodeling, which enhances cases of arrhythmogenesis and the propensity of sudden cardiac death [18,52]. Mapping ventricular fibrillation (VF) reveals various mechanisms ranging from wavelets of different stimuli to reentrant spiral waves [53]. 

Structural and electrical remodeling within a heart failure substrate can lead to increased arrhythmogenicity and VF [54]. This is in contrast to structurally normal hearts with VF characterized by unstable reentrant spiral waves or multiple waves and transient, highly dominant frequency regions [53,55]. Huang et al. showed that VF in heart failure is slower to activate, is more frequently blocked, and has fewer reentrant wave fronts compared to a structurally normal heart [56]. 

Apamin has been shown to heterogeneously increase the action potential duration (APD) within a heart failure substrate in both human studies with transplant hearts [57,58] and in animal models of heart failure [59,60]. Some studies have demonstrated that VF duration is decreased after apamin treatment [18,60,61]. Apamin treatment to the heart shortens VF duration and promotes its spontaneous termination [60]. Bonilla et al. [62] and Ni et al. [63] reported that apamin significantly prolonged APD in failing human and canine ventricular cardiomyocytes, along with the increased expression of SK channel protein in failing ventricles [18]. Treatment with apamin has been shown to decrease phase singularities and dominant frequencies during VF in a rabbit heart failure model [60]. Therefore, it has been suggested that an increase in myocyte contraction time could be a consequence of a more prolonged calcium entry due to the prolonged APD. This would increase myocyte energy needs, which is not good for a failing heart.

### 2.4. Liver Fibrosis

Liver fibrosis is caused by various types of chronic liver damage, which can be caused by chronic liver injuries, including viral agents, alcoholic hepatitis, and autoimmune hepatitis [64]. Cholestasis leads to hepatic accumulation of liver cytotoxic bile acids and inflammation of the liver, followed by biliary fibrosis, cirrhosis, and end stage liver disease [65,66]. Cholestatic liver disease, such as primary biliary cirrhosis and primary sclerotic cholangitis, is characterized by the gradual destruction of biliary epithelial cells and autoimmune and inflammatory diseases [67,68]. This progressive pathological process is described as the accumulation of extracellular matrix (ECM) proteins inside and around damaged liver tissue [69]. 

The process of liver fibrosis involves multiple cellular and molecular mechanisms in most chronic liver diseases. In addition, this process affects not only hepatocytes but also nonparenchymal cells, such as hepatic stellate cells (HSCs) and hepatic myofibroblasts, which are essential for maintaining liver structure and function [70]. Activated HSCs and portal fibroblasts enhance collagen deposition and act as the major cellular effectors of liver fibrosis [69,71]. Hepatocytes can also be transdifferentiated into mesenchymal cells through the deposition of collagen in the liver during epithelial mesenchymal transition (EMT) and chronic damage [72]. 

The cellular mechanism and antifibrotic effect of TGF-β1-induced hepatic fibrosis due to apamin have been explored [14]. Apamin prevents carbon tetrachloride-induced liver fibrosis [19]. Apamin treatment has led to decreased liver injury and pro-inflammatory cytokine levels. Treatment with apamin has resulted in a considerable reduction in the expression of TGF-β1, collagen I, fibronectin, and α-smooth muscle actin by suppressed Smad-signaling pathway TGF-β1-induced HSCs [19]. In addition, apamin has suppressed the activation of HSCs and the proliferation of biliary epithelial cells (BECs). Apamin has significantly inhibited bile duct proliferation and has reduced ECM accumulation in 3,5-diethoxycarbonyl-1,4-dihydrocollidine (DDC)-fed mice [19]. 

Kim et al. [19] suggested that apamin suppresses the proliferation of BECs and the activation of HSCs by inhibiting the TGF-β1 signaling pathway in hepatic fibrosis. In addition, some studies were conducted to investigate the antifibrosis or anti-epithelial mesenchymal transition (anti-EMT) mechanism by examining the effect of apamin on TGF-β1-induced hepatic fibrosis [14,19]. The anti-fibrotic effects of apamin in the carbon tetrachloride (CCl)4-induced liver fibrosis animal model were also examined in these studies. The major findings are that apamin alleviates the manifestation of liver tissue lesions and reduces the expression of TGF-β1 and fibronectin associated with liver fibrosis. Furthermore, other studies showed that apamin suppresses PDGF-BB-induced vascular smooth muscle cell proliferation and TGF-β1-induced hepatocyte EMT [14,73]. 

Treatment with TGF-β1 in AML12 murine hepatocytes results in losses of the E-cadherin protein at the cell–cell junctions and in an increased expression of vimentin [14]. Furthermore, the phosphorylation levels of Smad2/3, Smad4, ERK1/2, and Akt by TGF-β1 stimulation were increased. On the other hand, cells treated simultaneously with TGF-β1 and apamin maintained localized expression of high levels of E-cadherin and did not show an increase in vimentin [14]. Collected data provide in vitro evidence that apamin prevents hepatic epithelial cells from transitioning to the mesenchymal-like phenotype in response to TGF-β1 [14]. These results prove the potential of apamin for the prevention of EMT progression in vivo and in vitro. In summary, numerous studies have demonstrated that apamin inhibits TGF-β1-induced hepatocyte EMT in vitro and inhibits CCl4-injected fibrosis in vivo. The administration of apamin markedly increases the expression of epithelial marker E-cadherin and reduces mesenchymal marker vimentin in TGF-β1-induced hepatic fibrosis [14,19,74].

In addition, fibrotic liver tissue from both human and animal models showed increased expression of SK channels compared to control [75,76], but the distribution of SK channels in human liver differed according to the degree of fibrosis [77]. In fibrotic livers, SK channels were mainly expressed in hepatocytes, and in cirrhotic human livers, SK channels were mainly over-expressed in hepatocytes of cirrhosis nodules [76]. Likewise, Moller et al. [76] confirmed that expression of the SK channel genes increased during liver fibrogenesis in CCL4-induced liver injury mice and rats. Furthermore, the expression of SK channels was also significantly increased in three rat models of liver fibrosis induced by high-fat diet, bile duct ligation, or thioacetamide [76]. Freise et al. [75] found that SK channel blockers other than apamin inhibited the proliferation of a rat hepatic stellate cells and inhibited the expression of profibrotic genes in primary rat hepatic stellate cells.

### 2.5. Pancreatitis

Acute pancreatitis (AP) is a sudden inflammatory disease of the pancreas. The mortality rate associated with the most severe form of AP, characterized by pancreatic necrosis, is 30%–40% [78,79]. AP is caused by damage or disruption of pancreatic acini by digestive enzymes, such as amylase and lipase, which can break down tissue and cell membranes [80]. The acinar cells then produce proinflammatory cytokines, such as IL-1, IL-6, and TNF-α, which can induce acinar cell necrosis and can promote local pancreatic damage [80,81]. 

Bae et al. [20] explored the possibility that apamin suppresses AP on acinal cells via SK channel regulation and the role of SK channels in AP in acinar cells. Bae et al. [20] demonstrated evidence that apamin inhibits the development of AP by cerulein, a well-known substance that stimulates smooth muscles and induces digestion. Treatment with apamin in mice reduces serum amylase, lipase, cytokine, and myeloid oxidase activities. Based on Bae et al.’s study [20], the inhibitory effects of apamin have been demonstrated through c-Jun N-terminal kinases (JNK) inactivation in a cerulein-induced AP model in vivo. The administration of apamin inhibits the development of cerulein-induced AP, reducing inflammation, edema, cytokine production, and neutrophil infiltration in the pancreas. Apamin does not inhibit ERK1/2, p38, nor NF-kB. These results may suggest that apamin can prevent AP through the inhibition of JNK activation. 

### 2.6. CNS Disease 

Apamin is a very powerful and selective antagonist of the SK channel. This SK channel subtype is involved in the mediation of slow hyperpolarization that occurs after a train of action potentials, which in turn regulate neuronal excitability [82]. As a result of the pharmacological blockade of this particular SK channel, the after-hyperpolarization is reduced, and neuronal re-excitability is increased. Apamin can cross the blood-brain barrier and is rich in certain high-affinity sites associated with SK channels in the brain [83]. However, the increased neuronal excitability would be close to the potential possibility to trigger a seizure by damaged neurons as well an increased excitability due to altered membrane permeabilities. Further studies are required for this relation. SK channels are primarily located in the central nervous system and are present at a high density in the cerebral cortex and limbic system, which is known to be involved in cognitive and mood processes [82]. Hypothalamic neurons, in addition to the inwardly rectified K+ channels, also express SK channels sensitive to antagonism by apamin. These currents underlie the after-hyperpolarization that is observed at the tail end of the action potential or a prolonged depolarizing stimulus. [84,85].

SK channels play an important role in the repetitive activity of neurons [86] and block many hyperpolarizing inhibitory effects, including alpha-adrenergic, cholinergic, purinergic, and neurotensin-induced relaxation [87,88]. These channels, activated only by an increase in intracellular Ca2+, contribute to regulating the excitability and function of many cell types, including neurons, epithelial cells, T-lymphocytes, and skeletal muscle cells [89]. It has also been reported that the SK channels have the potential to inhibit inflammatory processes mediated by microglial cells [90,91,92].

The blockade of SK channels present in microglial cells is clearly mediated by the anti-inflammatory effects of apamin [91]. The blockage of SK channels by apamin increases the membrane potential of postsynaptic cells and increases the amplitude of excitatory postsynaptic potential with long-term effects [30,93]. Apamin not only protects the unmarred neuron but also restores the function of the silent neurons [3]. Therefore, the experimental studies on apamin have mainly focused on its use as an SK channel blocker in the central nervous system.

#### 2.6.1. Alzheimer’s Disease

Alzheimer’s disease (AD) is a neurodegenerative disorder affecting more than 40 million people worldwide and is expected to increase exponentially in the coming decades. The main cause of neurodegeneration in the brain affected by Alzheimer’s disease is due to the deposition of senile plaques, and the neurofibrillary tangles formed by tau proteins that accumulate in neuropil from the cerebral cortex and hippocampus [31]. 

An attention deficit is recognized as an early symptom of Alzheimer’s disease, which affects other cognitive areas and cellular mechanisms. The root cause of this attention deficit is being actively studied [94,95,96,97,98]. 

The early attention deficits are evident in TgCRND8 mice according to a well-established murine model of Alzheimer’s disease that recreates various features of the disease [99,100,101]. Attention control is performed by cholinergic regulation of the prefrontal cortex [102,103,104]. In addition, SK channels can be activated directly by Ca2+ influx through nicotinic receptors [105,106].

Proulx et al.’s study [107] showed that normal, robust cholinergic activation of the prefrontal attention circuitry was impaired in TgCRND8 mice and that this deficit can be pharmacologically corrected by inhibiting SK channels with the selective blocker apamin. Proulx et al. [16] confirmed that blocking SK channels through apamin treatment improved the efficiency of nicotinic signaling in layer 6 of the prefrontal cortex in TgCRND8 mice, according to an aggressive early onset model of brain amyloidosis. Studies have shown that apamin enhancement appears to be related to both the degree and timing of cholinergic attention regulation in the prefrontal cortex [102]. In addition, nicotinic receptor-mediated excitation severely damages the brain in Alzheimer’s disease mice, and the damage is sensitive to an SK channel blockade caused by apamin [108,109]. Accordingly, it has been demonstrated that apamin can overcome the inhibitory regulation of nicotinic receptors in the attention circuit of the frontal cortex [16]. 

Several behavioral and electrophysiological studies have suggested apamin for the treatment of AD, indicating that blocking of SK channels by apamin may enhance neuronal excitability, synaptic plasticity and organ long-term potentiation in the CA1 hippocampal region [31,110]. Apamin is considered useful for investigating physiological mechanisms related to brain functions, such as cognitive processes or mood control [31]. Thus, apamin has been presented as a new treatment strategy with added potential benefits in the treatment and improvement of the attention deficit in Alzheimer’s disease [111].

#### 2.6.2. Parkinson’s Disease

Parkinson’s disease (PD) is a neurodegenerative disorder characterized by typical motor symptoms due to the gradual loss of dopaminergic (DA) neurons in the substantia nigra, and it represents the second most common degenerative disease of the central nervous system [112]. Bradykinesia, rigidity, and resting tremors are highlighted as the main signs of PD. Although symptomatic treatment can provide benefits for many years, the disorder progresses slowly, eventually leading to a significant disability [113]. To carry out experiments, some studies used a chronic mouse model of MPTP intoxication that closely mimics the progressive loss of DA neurons in PD. Intoxication of the MPTP/probenecid paradigm represents a chronic, progressive, toxin-based mouse model of PD with substantial loss of DA neurons [114,115].

Previous studies suggest that bee venom can protect DA neurons from degeneration in experimental PD [12]. Interestingly, two studies have reported that peptide apamin can protect DA neurons in a model system of a midbrain culture that mimics the selective disappearance of these neurons in PD [116,117]. In some studies, it has been reported that two SK channel subtypes, SK2 and SK3, which are primarily blocked by apamin, exist in the neurons, resulting in a structural effect through their direct action on DA neurons [118,119,120]. The protective effect of apamin is due to the small excitatory increase in DA neurons that causes a moderate and sustained elevation of cytosolic calcium [116]. This is consistent with the pharmacological properties of apamin, which are known as potent and irreversible blockers of SK channels. These channels link the intracellular calcium transients to changes in membrane potential by promoting K+ efflux with increasing intracellular calcium during their action potential [121]. In addition, the use of apamin has been patented to overcome the shortcomings of drugs used in the treatment of PD, namely L-Dopa. 

As mentioned by Aufschnaiter et al. [31], SK channels control the firing frequency of neurons, especially at NMDA glutamatergic synapses, and are responsible for hyperpolarization following action potentials. Positive regulation in these channels promotes channel activity, impairing memory and learning, while negative regulation improves memory and learning, and reduces calcium channel sensitivity [122]. In neurons, this SK channel blockage decreases hyperpolarizing effects, regulating synaptic plasticity and memory encoding [31,123].

It has already been found that apamin has a protective effect against dopamine neurons in vitro [12,124]. It has been claimed that degenerative brain disease can be treated with an active ingredient consisting of an apamin component and other compounds for the treatment of PD [125].

### 2.7. Neurofibromatosis

Neurofibromatosis 1 (NF1) is a common genetic disorder that affects about 1 in 3500 people [126]. The disease is characterized by a number of physiological symptoms, including benign and malignant tumors of the nerve sheath and the CNS, hyperpigmentation of the skin, irises, and hippocampus, unidentified bright objects in the brain and macroencephaly [127]. In addition, about 30%–60% of NF1 patients suffer from some type of learning deficit [128]. Patients with NF1 experience difficulties in both verbal and nonverbal learning tasks [129,130], and have been reported to have particular difficulties in working with visuospatial tasks [131,132,133].

Kallarackal et al. [21] found that Nf^+/-^ mice in genetically engineered murine models with heterozygous mutations in Nf1 have a spatial cognitive deficit. The learning deficits in Nf1^+/−^ mice are similar to those observed in humans and tend to be primarily visuospatial. In addition, very similar to the human state, learning deficits occurring in Nf1+/- mice occur in approximately 40%–60% of mice with mutations [134]. 

SK channels are associated with the possible role of learning regulation and dopamine-related disorders. SK channels are believed to affect learning by regulating synaptic plasticity through NMDA receptors [30,135]; however, the defect observed in Nf1^+/-^ mice was confirmed to have been eliminated by treatment with apamin (either through acute intraperitoneal injections or chronic micro-pump delivery) [21]. 

Blockage of SK channels by apamin increases membrane potential in postsynaptic cells and increases the amplitude of excitatory postsynaptic potential as a result of long-term potentiation [30,93]. These data suggest that apamin may be an effective treatment for option learning disorders that potentially appear in Nf1^+/-^ mice and patients. In addition, early studies have shown that systemic apamin administration facilitates learning and memory [3]. Several studies have highlighted the relevance of the SK channels in information processing and storage at the system level [3,21,136]. These studies suggest that SK channels are an appropriate target for apamin treatment for learning deficits.

### 2.8. Atopic Dermatitis

Inflammatory skin diseases such as atopic dermatitis involve the increased infiltration of some inflammatory cells, such as lymphocytes, macrophages, some eosinophils, and dendritic cells [137]. These inflammatory cells secrete various inflammatory cytokines, such as IL-1, TNF-α, interferon (IFN)-γ, and IL-6, after stimulation [138]. Increased inflammatory cytokines play an important role in the pathogenesis of atopic dermatitis [139,140]. In particular, TNF-α and IFN-γ can induce type-2 T helper cell (Th2)-related chemokines, as well as activation of regulating chemokines and macrophage-derived chemokines [141]. These Th2-related chemokines are considered to play an important role in developing atopic dermatitis [139,142]. Many researchers have attempted to inhibit the inflammatory cytokines and chemokines in TNF-α- and IFN-γ-stimulated keratinocytes and have analyzed the biological and pharmacological mechanisms with various factors [143,144,145].

Kim et al. [15] investigated the anti-inflammatory effect of apamin on TNF-α- and IFN-γ-induced inflammatory response in human keratinocytes. Apamin administration prevents the activation of JAK/STAT and NF-κB, which are transcription factors associated with inflammatory cytokines in TNF-α- and IFN-γ-treated human keratinocytes. These results showed that apamin has anti-inflammatory effects on atopic dermatitis. 

Apamin improves inflammatory conditions through the inhibition of Th2-related cytokines [15]. In addition, apamin down-regulates the activation of the JAK/STAT and NF-kB signaling pathways in human keratinocytes. The expressions of inflammatory cytokines, such as IL-1β and IL-6, and Th2-related chemokines, including thymus- and activation-regulated chemokines and macrophage-derived chemokines, are increased in TNF-α-and IFN-γ-stimulated human keratinocytes. Apamin has an inhibitory effect on TNF-α- and IFN-γ-induced activation of the JAK/STAT and NF-κB signaling pathways [15]. Therefore, these results suggest that apamin has a therapeutic effect on atopic dermatitis through the amelioration of inflammatory conditions.

## 3. Conclusions

Apamin is currently being researched to target and to develop new therapeutic agents. In this review, its potential therapeutic and pharmacologic applications for non-neoplastic diseases have been discussed, and the focus has been the emerging roles and the functions of apamin in the pathogenesis of inflammation and fibrosis-related diseases as a novel regulatory agent. In addition, further studies are required to examine the toxicity, including experimental descriptions of optimal doses, allergic reactions, and side effects. The evaluation of pharmacokinetics and biological properties are also important. Collectively, therapy with apamin could be a potential therapeutic alternative for the treatment of non-neoplastic diseases; however, more research is needed to examine the toxicity and the pharmacokinetics for realistic treatment and applications. 

## Figures and Tables

**Table 1 toxins-12-00195-t001:** Pharmacological actions of apamin for various diseases.

Disease Entity	Experimental Model	Biological Role	Molecular Mechanisms	Reference
Atherosclerosis	- THP-1 cell treated with oxLDL - LPS injection with high fat diet	Inhibited apoptosis	Decreased NF-κB signaling pathway	[17]
Heart failure	Pacing-induced heart failure	Increased the action potential duration	SK channel blockade	[18]
Liver fibrosis	- AML12 cell treated with TGF-β1 - DDC-fed or CCl_4_-injection mice	Suppressed hepatic fibrosis	Inhibited MAPK, Smad, and TGF-β1 signaling pathway	[14,19]
Pancreatitis	Cerulein-injected mice	Attenuated cytokine production	Suppressed JNK activation	[20]
Alzheimer’s disease	Transgenic mice	Improved memory acquisition	Improved efficiency of nicotinic signaling	[16]
Parkinson’s disease	MPTP/probenecid-injection PD mice	Hypercholinergic state to DA denervation	SK channel blockade	[12]
Neurofibromatosis	Heterozygous Nf1^+/−^ mouse model	Increased membrane potential in postsynaptic cell	SK channel blockade	[21]
Atopic dermatitis	HaCaT cell treated with TNF-α/IFN-γ	Suppressed inflammatory cytokines	Inhibited JAK/STAT and NF-κB signaling pathway	[15]

THP-1, human monocytic cell line; oxLDL, oxidized low-density lipoprotein; LPS, lipopolysaccharide; DDC, 3,5-diethoxycarbonyl-1,4-dihydrocollidine; CCl_4_, carbon thtrachloride_4_; AP, acute pancreatitis; PD, Parkinson’s disease; SK, small conductance Ca^2+^-activated K^+^; MPTP, 1-methyl-4-phenyl-1,2,3,6-tetrahydropyridine; HaCaT, spontaneously transformed aneuploid immortal keratinocyte cell line.

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
