# Peer review of "Therapeutic Effects of Apamin as a Bee Venom Component for Non-Neoplastic Disease"

_toxins, 2020, doi:10.3390/toxins12030195_

Round 1
Reviewer 1 Report
In this review article, the authors discussed the therapeutic and pharmacological potential of apamin, a bee venom derivative, for inflammatory diseases. Overall, this topic is timely and interesting, and the authors have done well summarizing both older and current works. The paper is well-balanced and comprehensive. I have only a few suggestions to strengthen the flow and impact of this review.
Line 55-57, Ras pathway is mentioned here.
The authors briefly described the requirement of Ras for the expression of SK channel here. It does not seem necessary to mention Ras here as there was no further discussion regarding Ras pathway across the manuscript. Does apamin have therapeutic effect on any symptoms related to rasopathy? If not, I suggest to remove the description of ras pathway in this sentence
Line 58 “…and they may play specific roles in dopamine-related disorders.”
Please provide a reference here.
Line 61 “…thereby contributing to learning and memory control.”
I recommend the authors to cite relevant references (if any) related to effect of apamin on the learning and memory control to support this sentence.
Line 271 “In some studies, it has been apamin reported that …”
I guess ‘apamin’ was miswrote here.
Reviewer 2 Report
I read the paper entitled:"Therapeutic effects of apamin as a bee venom component for inflammatory disease" submitted for publication to Toxins.
The paper reviews the current understanding of apamin, a neurotoxin contained in bee venom, in the treatment of apoptosis, fibrosis, and central nervous system diseases.
Apamin is a well known toxin in science, especially in the field of electrophysiology where it is utilized since many years as a specific blocker of SK subtype K channels. Thus the neurophysiological effects of this toxin are well undestood and analysed using direct fine electrophysiological techniques like voltage-clamp and patch-clamp.
There are also many reviews (published also in this journal) about the effect of bee venom components (including apamin) in different pathological processes, including cancer. After a brief check of the literature on the topic I found papers treating similar or very similar issues which are not cited at all in this review, despite they are published on this same journal and very recently. For instance, just briefly I point out a paper by Silva J et al in 2015 on Toxins entitled "Pharmacological Alternatives for the Treatment of Neurodegenerative Disorders: Wasp and Bee Venoms and Their Components as New Neuroactive Tools" or a very recent paper by Aufschneiter et al entitled "Apitoxin and Its Components against Cancer, Neurodegeneration and Rheumatoid Arthritis: Limitations and Possibilities" always published on Toxins in 2020. Both these papers include arguments treated in the submitted manuscript and they should be mentioned and discussed.
Therefore I feel that the field is already well informed about bee venoms, their pharmacological properties and their possible therapeutical use.
But even if we can consider the possibility that, in any case, there should be space for another review on the argument, I find the submitted review a little bit confused with the respectable aim to combine research data and clinical and medical data, but, at the end, failing to reach this target.
If we consider the title, for instance, I find difficult to consider Alzheimer's and Parkinson's as an "inflammatory disease". Better, in this topic, multiple sclerosis, which, indeed, is considered as a possible therapeutical target of bee venoms. Moreover, in one of the cited papers about the role of apamine on dopaminergic neurons degeneration, apamine efficacy (which is qualitatively similar, but lower than veratridine, a Na channel blocker) is considered in case of Dopamine neurons increased excitability during hypoxia or ischemic consitions (a possible role of free radicals is still considered in Parkinson's, but we are far form an inflammatory state).
The same criticisms for neurofibromatosis: apamin treatment regards specific neurophysiological aspects (learning and synaptic plasticity) and not, as suggested by the title, inflammatory issues. Moreover, I have the same concerns as above regarding to consider this disease as an "inflammatory" disease, instead of a genetic disease with glial tumors and an altered nervous system development.
Just below a completely different disease is considered. Atopic dermatitis, an inflammatory disease where apamin seems to act on different singalling basis.
Above, on the contrary, again something that is not related with an inflammatory disease: ventricolar fibrillation. A true electrical or electrophysiological problem where apamin deserves a special attention for its well known slective specific blocking effect on SK channels.
In conclusion I think that this MS needs profound revisions, firstly by choosing an appropriate sequence of arguments which should be logically related to the title and to the aim expressed in the title.
For this reason I suggest to reject this paper
Reviewer 3 Report
This review must to be rewritten in a better way. It's sounds confuse. Some parts were written in a wrong way. In Table 1 the references were missing. The 2.2, 2.3 and 2.4 can be written together because the information is the same.
Reviewer 4 Report
The manuscript entitled "Therapeutic effects of apamin as a bee venom component for inflammatory disease" described the therapeutic application of bee venom, apamin. Bee venom is one of the promising alternative medicine. This paper introduces the readers the scientific background of bee venom for therapeutic purpose. Some revisions are necessary to improve the readability.
(1) As far as I know, there are three subtypes of SK-channels, and apamin has the different activities against SK1, SK2, and SK3. Please explain this in section 2.1.
(2) According to the description in the main body, apamin down-regulate the NF-κB signal pathway (line 120). But the description of molecular mechanism in table 1, apamin increase NF-κB. I could not understand this inconsistency very well.
(3) I know the expression and sensitivity of SK-channels are increased in animal models of heart failure and so on. But I could not understand the sentence in lines 145-146, because the Bonilla and Ni did not use the failing ventricles “due to treatment with apamin”. Please correct this sentence properly. And according to the table 1, “apamin up-regulate SK-channels”, I think “SK channel blockade” is suitable here.
(4) Please indicate that liver express SK channels. And if you know SK channel’s distribution in liver (in parenchymal or non parenchymal cells?), please discuss about this in section 2.6.
(5) Some of the TNF-alpha are replaced to “TNF-a”, not alpha.
(6) Please select whether “afterhyperpolarization” or “after-hyperpolarization”
(7) Please indicate the complete expression of “BEC” (biliary epithelial cells?).
Round 2
Reviewer 2 Report
I read the revised version of the MS # toxins-725977 submitted for evaluation to this Journal.
The paper is considerably improved with respect to the previous version. The different paragraphs are better developed making clear the purpose of the authors to give reader a wide spectrum review of apamin potential effects as a therapeutical agent in very different fields of medicine and different diseases.
I still have some concerns about a deeper discussion of the potential apamin effects on heart and central nervous system due to its electrophysiological properties.
For instance, if apamin is active in reducing VF by increasing myocyte Action potential duration (APD), then I would expect an increase in myocyte contraction time, as a consequence of a more prolonged calcium entry (due to the prolonged APD). This last aspect would increase myocyte energy need, a factor not properly good for a failing heart. It would perhaps be necessary adding some words discussing this, in my opinion, discrepancy.
At the level of central nervous system, apamin should increase neuronal excitability, by decreasing AP hyperpolarization phase. The potential neurophysiological consequencies of this effect are clearly explained. However this increased neuronal excitability should also get closer to the potential possibility to trigger in an easier way a seizure, especially if we are talking about nervous system pathological alterations, where there is glutamate release by damaged neurons as well an increased excitability due to altered membrane permeabilties. Also here, I would add some words discussing the possibility or not of this possibility.
I think that now the paper is suitable for publication after minor revision as suggested and obviously according to the editor choice.
Author Response
Reviewer #2
#2: comments 1
I still have some concerns about a deeper discussion of the potential apamin effects on heart and central nervous system due to its electrophysiological properties.
For instance, if apamin is active in reducing VF by increasing myocyte Action potential duration (APD), then I would expect an increase in myocyte contraction time, as a consequence of a more prolonged calcium entry (due to the prolonged APD). This last aspect would increase myocyte energy need, a factor not properly good for a failing heart. It would perhaps be necessary adding some words discussing this, in my opinion, discrepancy.
Answer: Thank you for your good comment. We add some word discussing your suggestion.
Revised manuscript:
2.3. Ventricular fibrillation
Therefore, it would be suggested that an increase in myocyte contraction time could be a consequence of a more prolonged calcium entry due to the prolonged APD. This would increase myocyte energy need which is not a good factor for a failing heart.
2: comments 2
At the level of central nervous system, apamin should increase neuronal excitability, by decreasing AP hyperpolarization phase. The potential neurophysiological consequencies of this effect are clearly explained. However this increased neuronal excitability should also get closer to the potential possibility to trigger in an easier way a seizure, especially if we are talking
about nervous system pathological alterations, where there is glutamate release by damaged neurons as well an increased excitability due to altered membrane permeabilties. Also here, I would add some words discussing the possibility or not of this possibility.
I think that now the paper is suitable for publication after minor revision as suggested and obviously according to the editor choice.
Answer: Thanks for your comment. We agree with your suggestion. We mentioned another some words discussing your comments.
Original manuscript:
2.6. CNS disease
Apamin can cross the blood-brain barrier and is rich in certain high-affinity sites for apamin associated with SK channels in the brain [84]. SK channels are primarily located in the central nervous system and are present at high density in the cerebral cortical and limbic system, which is known to be involved in cognitive and mood processes [83].
Revised manuscript:
2.6. CNS disease
Apamin can cross the blood-brain barrier and is rich in certain high-affinity sites for apamin associated with SK channels in the brain [84]. However, the increased neuronal excitability would be close to the potential possibility to trigger a seizure by damaged neurons as well an increased excitability due to altered membrane permeabilities. Further studies are required for this relation. SK channels are primarily located in the central nervous system and are present at high density in the cerebral cortical and limbic system, which is known to be
involved in cognitive and mood processes [83].
Reviewer 3 Report
Dear authors,
I revised the manuscript Therapeutic effects of apamin as a bee venom 2 component for non-neoplastic disease.
This version is better than the last one. It sounds comprehensive now. Some words must to be checked.
Take a look at the section Atopic dermatitis because keratinocytes is not infiltrated inflammatory cells (first paragraph third line).
Regards.
Author Response
Reviewer #3
Reviewer #3: comments 1
I revised the manuscript Therapeutic effects of apamin as a bee venom 2 component for nonneoplastic disease.
This version is better than the last one. It sounds comprehensive now. Some words must to be checked.
Take a look at the section Atopic dermatitis because keratinocyte is not infiltrated inflammatory cells (first paragraph third line).
Answer: Thank you for your kind comment. We modified the Atopic dermatitis section as your comments.
Original manuscript:
2.8. Atopic dermatitis
These keratinocytes and inflammatory cells secrete various inflammatory cytokines, such as IL-1, TNF-α, interferon (IFN)-γ, and IL-6, after stimulation [139].
Revised manuscript:
2.8. Atopic dermatitis
These inflammatory cells secrete various inflammatory cytokines, such as IL-1, TNF-α, interferon (IFN)-γ, and IL-6, after stimulation [139].